# Hepatitis E Virus Quasispecies in Cerebrospinal Fluid with Neurological Manifestations

**DOI:** 10.3390/vaccines9101205

**Published:** 2021-10-19

**Authors:** Florence Abravanel, Florence Nicot, Sébastien Lhomme, Michele Cazabat, Thomas Drumel, Aurélie Velay, Justine Latour, Julie Belliere, Pascal Cintas, Nassim Kamar, Jacques Izopet

**Affiliations:** 1INSERM, Insitut Toulousain des Maladies Infectieuses et Inflammatoires (INFINITY), INSERM URN1291-CNRS UMR 5051, Université Toulouse III, F-31300 Toulouse, France; lhomme.s@chutoulouse.fr (S.L.); cazabat.m@chu-toulouse.fr (M.C.); kamar.n@chu-toulouse.fr (N.K.); izopet.j@chu-toulouse.fr (J.I.); 2CHU Toulouse, Hôpital Purpan, Laboratoire de Virologie, National Reference Center for Hepatitis E, F-31300 Toulouse, France; nicot.f@chu-toulouse.fr (F.N.); latour.j@chu-toulouse.fr (J.L.); 3CHU Nantes, Laboratoire de Virologie, CEDEX 1, F-44093 Nantes, France; thomas.drumel@chu-nantes.fr; 4CHU Strasbourg, Laboratoire de Virologie, INSERM, IRM UMR 1109, F-67000 Strasbourg, France; aurelie.velay@chru-strasbourg.fr; 5CHU Toulouse, Hôpital Rangueil, Département de Néphrologie, Dialyse et Transplantation Multiorgane, F-31300 Toulouse, France; belliere.j@chu-toulouse.fr; 6INSERM, U1048, Institut des Maladies Métaboliques et Cardiovasculaires, Rangueil, F-31300 Toulouse, France; 7CHU Toulouse, Hôpital Purpan, Département de Neurologie, F-31300 Toulouse, France; cintas.p@chu-toulouse.fr

**Keywords:** hepatitis E virus, quasispecies, chronic infection, cerebrospinal fluid, compartmentalization

## Abstract

Hepatitis E virus (HEV) infection can lead to a variety of neurological disorders. While HEV RNA is known to be present in the central nervous system, HEV quasispecies in serum and cerebrospinal fluid (CSF) have rarely been explored. We studied the virus’ quasispecies in the blood and the CSF of five patients at the onset of their neurological symptoms. The samples of three patients suffering from meningitis, neuralgic amyotrophy and acute inflammatory polyradiculoneuropathy were taken at the acute phase of the HEV infection. The samples from the other two patients were taken during the chronic phase (5 years after HEV diagnosis) when they presented with clinical signs of encephalitis. We sequenced at least 20 randomly polyproline regions of the selected virus clones. Phylogenetic analysis of the virus variants in the blood and the CSF revealed no virus compartmentalization for the three acute-phase patients but there was clear evidence of HEV quasispecies compartmentalization in the CSF of the two patients during chronic infection. In conclusion, prolonged infection in the immunocompromised condition can lead to independent virus replication in the liver and the tissues, producing viruses in CSF.

## 1. Introduction

HEV is the most common cause of hepatitis worldwide. It was originally thought to be a disease of developing countries, where genotypes 1 and 2 predominate and where it is transmitted by ingestion of fecally contaminated water [1]. The virus is also endemic in high-income countries, including European countries, where genotypes 3 and 4 predominate; it is mostly transmitted by porcine zoonosis [1]. The HEV genome, a single-strand approximately 7.2 kb-long positive-sense RNA, contains three open reading frames (ORFs), ORF1, ORF2, and ORF3, flanked by noncoding regions. ORF1 encodes a nonstructural protein with at least four putative functional domains: methyltransferase, papain-like cysteine protease, helicase, and RNA-dependent RNA polymerase. It also has domains that are homologous with those of other plant and animal positive-strand RNA viruses: Y domain, polyproline region (PPR) or hypervariable region (HVR), and a macro domain (X domain). ORF2 encodes the capsid protein, while ORF3 encodes a small phosphoprotein that is involved in virus egress [1]. Host sequences have been found incorporated into the PPR of the HEV genome of infected patients [2,3,4]. In vitro studies have shown that these insertions foster virus growth in vitro and may also alter cell tropism [5]. HEV exists as a quasispecies, a mixture of closely related variants, like all RNA viruses [2,6]. This heterogeneity is a factor in HEV pathogenesis and virus adaptation [7,8,9].

Most patients with an acute hepatitis E infection have no symptoms, while the symptoms of an acute infection are indistinguishable from those of the hepatitis caused by other acute viral infections [1]. HEV replicates non-cytopathically so that its pathology is believed to be immune-mediated. HEV genotype 3, 4, and 7 infections are responsible for chronic hepatitis in immunocompromised patients: solid organ transplant patients, patients with hematological malignancies, or those with a low CD4 cell count due to HIV [1]. An infection results in a spectrum of extrahepatic manifestations, including renal disease, pancreatitis, and a variety of neurological disorders [10]. Clinical cohort studies also revealed neuralgic amyotrophy (Parsonage–Turner syndrome), Guillain–Barre syndrome, and encephalitis [10,11]. These neurological manifestations may be caused directly or indirectly [10]. HEV replication may directly affect tissues, causing local tissue damage and inflammation [12], while cross-reactive immune responses during or after infection may also indirectly damage tissues. HEV RNA has been found in the cerebrospinal fluid (CSF) of some but not all patients with neurological manifestations [10]. The presence of different HEV quasispecies in a patient’s serum and CSF suggests that neurotropic HEV variants have emerged [13].

We aimed to investigate virus diversity in the serum and the CSF in five patients with neurological manifestations observed during the acute or the chronic phase of HEV infection.

## 2. Materials and Methods

### 2.1. Patients

The five HEV-infected patients with neurological manifestations all had HEV RNA in their blood and CSF samples (Table 1). They were all infected with HEV genotype 3. These samples were collected when the neurological manifestations occurred. Two immunocompetent patients had an acute hepatitis E infection and the three other patients were immunocompromised: one heart transplant recipient, one kidney transplant recipient, and one patient with a low CD4 cell count (89/mm^3^) due to HIV. The clinical manifestations of two patients were described previously [14,15]. The two patients had been infected chronically with HEV for 5 years when the blood and the CSF were sampled. Tests were performed retrospectively on the blood and CSF samples collected in France. This was a noninterventional study. The biological materials and the clinical data were obtained only via standard viral diagnostics following a physician’s order (no supplemental or modified sampling). The data were analyzed anonymously. According to the French law (Loi Jardé), anonymous retrospective studies do not require institutional review board approval.

### 2.2. Cloning and Sequencing of the PPR

Virus RNA was isolated from the blood and CSF samples using a QIAamp Viral RNA kit (Qiagen, Courtaboeuf, France) and a fragment covering the PPR was amplified by reverse transcription PCR (RT-PCR). A nested reverse transcriptase PCR (RT-PCR) was used to amplify a 325 nt fragment of the HEV genome. RT-PCR was performed with a SuperScript III One-Step RT-PCR system (Invitrogen, Cergy-Pontoise, France) under the following conditions: 30 min at 50 °C, 2 min at 94 °C, and 50 cycles of 30 s at 98 °C, 30 s at 55 °C, and 35 s at 72 °C. The sense primer was 1710-S (GAGTGCCGYACKGTGCTYGGGAATAA) and the antisense primer was 3050-AS (ACATCRACATCCCCCTGYTGTATRGA). Then, a nested PCR was performed with an Expand High Fidelity PCR system (Roche, Mannheim, Germany) under the following conditions: 30 s at 98 °C and 30 cycles of 30 s at 98 °C, 30 s at 55 °C, and 30 s at 72 °C. The sense primer was IS-G3 (ACCYTGTAYACTCGNACCTGGTC) and the antisense primer was IA-G3 (ACCTTRGCRCCGTCAGGRTAGGT).

The purified PCR products were quantified by means of spectrophotometry. We then directly ligated 10 ng cDNA into 10 ng of TOPO XL-2 (Complete PCR Cloning Kit, Invitrogen, Cergy-Pontoise, France). Recombinant plasmids were used to transform *Escherichia coli* competent cells (Invitrogen, Cergy-Pontoise, France) according to the manufacturer’s protocol and the resulting transformants were grown on ampicillin plates. At least 20 clones were randomly selected for sequencing. PCR products were sequenced on both strands using the dideoxy chain termination method (PRISM Ready Reaction AmpliTaq Fs and BigDye Terminator; Applied Biosystems, Paris, France) on an ABI 3130XL analyzer (Applied Biosystems, Foster City, CA, USA).

We used the MAFFT version 7.45 sequence alignment tool followed by IQTREE version 1.6.12 to construct a bootstrapped tree (1000 replicates) to obtain the maximum likelihood model. Finally, Interactive Tree Of Life (iTOL) v3 was used to visualize the tree [16]. If several clones had similar sequences in the compartment (blood or CSF), only one sequence was maintained in the tree.

## 3. Results

We studied the virus quasispecies of three patients at the acute phase of HEV infection presenting with different neurological manifestations: meningitis (patient 1), neuralgic amyotrophy (patient 2), and acute inflammatory polyradiculoneuropathy (patient 3) (Table 1). For these three subjects (patients 1, 2, and 3), the phylogenetic analysis did not indicate virus compartmentalization between the blood and the CSF (Figure 1), the blood and the CSF clones were indistinguishable. Two patients recovered without treatment while patient 3 died from multiorgan failure on day 153 post-transplantation.

We also investigated the virus quasispecies in the blood and the CSF of the two chronically infected (5 years) HEV patients who presented with clinical signs of encephalitis (patients 4 and 5) (Table 1). Phylogenetic analysis revealed marked compartmentalization of the HEV quasispecies in their CSF. In one case, the HEV infection of one (patient 4) was successfully treated with ribavirin, and the patient recovered; the other patient was noncompliant and was lost to follow-up.

The phylogenetic tree for patient 4 quasispecies showed distinct clusters (Figure 1): one including sequences from the variants detected only in the plasma, another cluster contained the variants present in both the blood and the CSF, and the third included only the CSF variants that had a 185 nt insertion in the PPR region. This insertion was a duplication of the PPR region.

The phylogenetic tree for patient 5 also revealed distinct populations of virus variants in the blood and the CSF (Figure 1). One cluster in the CSF contained variants with an 87 nt insertion in the PPR region with a sequence that was a duplication of the HEV PPR region. This suggests that there was HEV quasispecies compartmentalization in both chronically HEV-infected patients with neurological manifestations.

## 4. Discussion

Several neurological manifestations associated with HEV infection have been described and in vitro studies suggest that HEV can replicate in the brain. Our comparison of the HEV quasispecies in the blood and the CSF of the patients with neurological manifestations indicates that there is no compartmentalization of HEV quasispecies in the CSF at the acute phase of infection and that it arises only during a chronic infection.

Close to half (42.9%) of patients with neurological manifestations can have HEV RNA in their CSF [17], and several neural cell lines [18], such as induced pluripotent stem cell-derived human neurons [17] and primary mouse neurons [17], are susceptible to HEV infection. A study on the Mongolian gerbil model revealed immunohistochemical staining of the HEV ORF2 protein mainly in the cytoplasm of neurons, ependymal epithelium cells, and the choroid plexus area [12]. Similarly, using a rabbit model, both positive- and negative-stranded HEV RNA and HEV antigen expression were detected in the liver and the brain of necropsied rabbits [19].

Viruses enter the brain by a variety of means, primarily via neuronal transport or by crossing one of the several barriers to the central nervous system, including the blood–brain barrier and the blood–cerebrospinal fluid barrier (choroid plexus) [20]. An in vitro study showed that primary cultures of the human brain’s microvascular cells infected with HEV lost their tight junction proteins, including claudin-5, occludin, and ZO-1 (zonula occludens-1) [21]. These findings suggest that HEV invades the CNS by disrupting the blood–brain barrier.

Our investigation of the HEV quasispecies in the blood and the CSF of the five patients with acute or chronic HEV infections and neurological manifestations found distinct HEV variants, indicating compartmentalization only in chronic-phase patients. This agrees well with a previous study that found virus compartmentalization in the CSF of a chronically HEV-infected kidney transplant recipient suffering from peripheral demyelinating polyradiculoneuropathy after 33 months of HEV infection [13]. A virus can enter a host as a diverse quasispecies but only a few variants may cross the host’s barriers or survive selective pressure. Those that overcome these barriers may replicate freely to give a low-diversity population until expansion and accumulated mutations restore diversity [22]. Our findings suggest that diversification requires prolonged replication in the central nervous system. Perhaps neurological manifestations appear after the emergence of neurotropic HEV variants. In contrast, the virus population in the central nervous system of the patients with acute hepatitis E and neurological symptoms did not have time to diversify. Interestingly, our results are in line with a recent study that has reported detection of HEV-RNA in seminal plasma and semen only in chronically HEV-infected patients. Genomic sequencing showed significant differences between viral strains in the ejaculate compared to stool, suggesting HEV replication in the male reproductive system during chronic hepatitis E [23].

The duplication of the virus’ PPR region in patients with clear compartmentalization of the HEV quasispecies in the CSF was detected only in the samples collected when the neurological symptoms occurred. The sequence encoding this region can vary in both composition and length, depending on the HEV subtype [4,7,8,24,25]. Recombinant strains of HEV with insertions of human genome fragments or HEV sequence duplications in the PPR have been found in both acute-phase and chronically infected patients [4]. These genomic rearrangements may increase the net protein load by increasing the proportion of positively charged amino acids or by decreasing the negatively charged ones [4]. Sequences with genomic rearrangements seem to have more potential new regulation sites: for ubiquitination, acetylation, and phosphorylation [4]. Whether this duplication promotes the neurotropism of the HEV virus requires further investigation.

## 5. Conclusions

In conclusion, compartmentalization of the virus in the central nervous system seems to require a prolonged infection. The cells that support HEV replication in the human central nervous system have yet to be identified.

## Figures and Tables

**Figure 1 vaccines-09-01205-f001:**
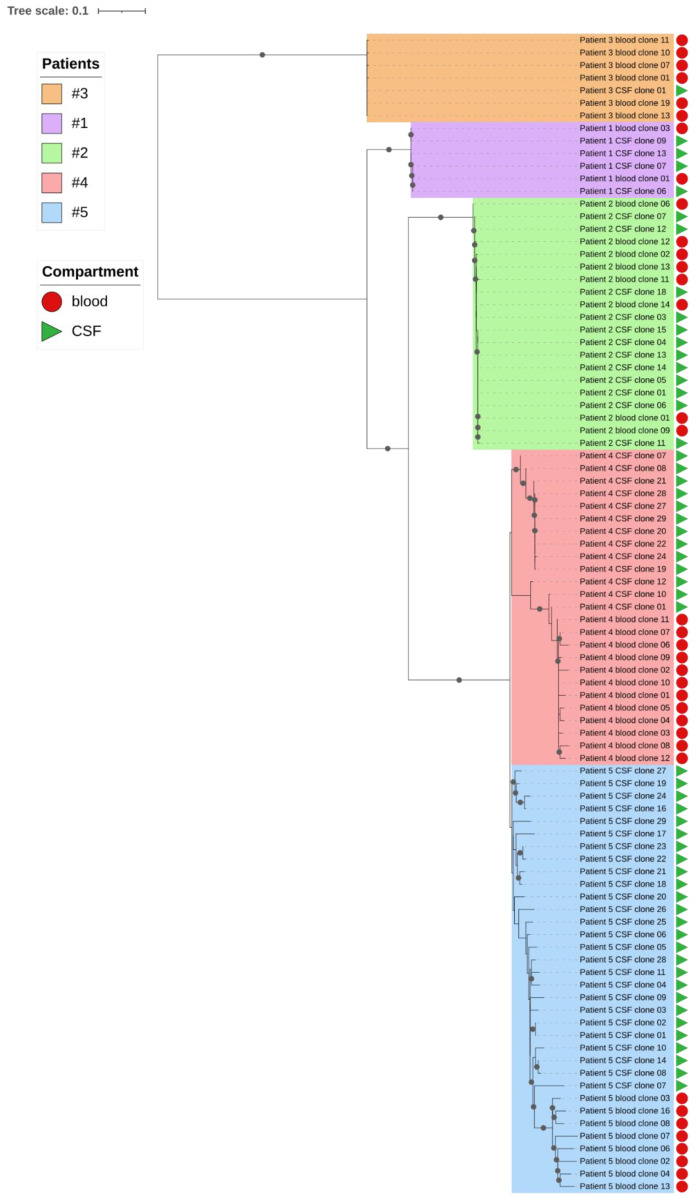
Phylogenetic analysis of the hepatitis E virus quasispecies in the blood and the cerebrospinal fluid of the five patients. The neighbor-joining tree was produced using the maximum likelihood distance clone sequences. The phylogenetic trees were visualized with iTOL. Black dots represent the nodes with bootstrap values >70.

**Table 1 vaccines-09-01205-t001:** Patients’ characteristics.

	Patient 1	Patient 2	Patient 3	Patient 4	Patient 5
Age	50	55	56	33	53
Sex	Male	Male	Male	Male	Female
Neurological manifestation	Meningitis	Parsonage–Turner syndrome	Acute inflammatory Polyradiculo-neuropathy	Encephalitis	Encephalitis
Comorbidities	Alcoholic cirrhosis	None	Heart transplantation	Kidney transplantation	HIV infection
ALT (IU/L, normal values < 35)	2975	1252	63	45	52
Plasma HEV-RNA log_10_ IU/mL	5	7.2	6.9	6	5.3
CSF HEV-RNA log_10_ IU/mL	2.9	3	2.4	3	5.7
CSF analysis: Protein (g/L) White blood cells/mL	0.42 0	0.44 4	0.54 20	1.03 16	0.49 6

## Data Availability

Raw data can be asked to the corresponding author.

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
