# Peer review of "Hepatitis E Virus Quasispecies in Cerebrospinal Fluid with Neurological Manifestations"

_vaccines, 2021, doi:10.3390/vaccines9101205_

Round 1

Reviewer 1 Report

Comments to the authors

The manuscript entitled “Hepatitis E virus quasispecies in the cerebrospinal fluid” investigated and compared HEV diversity in serum and cerebrospinal fluid in three acutely and two chronically infected patients with neurological manifestations. The study is very interesting and adds more knowledges about HEV-related neurological disorders at the virological level. The following minor issues should be addressed prior to the acceptance of the article.

  1. Have the virus strains sequenced? If so, what are the genotypes of the virus strains infected in the five patients?
  2. In the method section 2.2, the length of the amplified fragment should be provided.
  3. It is suggested to modify Figure 1 to be more suitable to be visualized. For example, the image aspect ratio is inappropriate at present.
  4. In line 10 of the result section, "table 2" instead of "table 1". And the table should change to Three-line table format.
  5. In line 1-2 of the discussion section, the statement lacks references. Besides, in vivo study such as in HEV-3ra-infected rabbits (Jian H. et.al., PLoS One, 2014), negative strand HEV RNA was also detected in the brain, indicating of active virus replication.
  6. In the discussion section, line 3-5 and line 19-21 (first sentence of the fourth paragraph) are describing same things repeatedly. It is recommended to delete some duplicate result descriptions to make the manuscript more concise.
  7. One last and small suggestion for you to be considered. It might be better to change the title to "Hepatitis E virus quasispecies in the cerebrospinal fluid of patients with neurological manifestations."

Author Response

  1. Have the virus strains sequenced? If so, what are the genotypes of the virus strains infected in the five patients?

All the patients were infected by HEV genotype 3. This is now mentioned in the method section

  1. In the method section 2.2, the length of the amplified fragment should be provided.

It is now provided in the method sectio (325 nt)

  1. It is suggested to modify Figure 1 to be more suitable to be visualized. For example, the image aspect ratio is inappropriate at present.

We have now provded a new Figure 1. We hope i twill be suitable for Vaccines journal.

  1. In line 10 of the result section, "table 2" instead of "table 1". And the table should change to Three-line table format.

We have changed the format of Table 1 in the revised manuscript.

  1. In line 1-2 of the discussion section, the statement lacks references. Besides, in vivo study such as in HEV-3ra-infected rabbits (Jian H. et.al., PLoS One, 2014), negative strand HEV RNA was also detected in the brain, indicating of active virus replication.

We have now added this reference in the discussion section.

  1. In the discussion section, line 3-5 and line 19-21 (first sentence of the fourth paragraph) are describing same things repeatedly. It is recommended to delete some duplicate result descriptions to make the manuscript more concise.

We have now modified this section as requested

  1. One last and small suggestion for you to be considered. It might be better to change the title to "Hepatitis E virus quasispecies in the cerebrospinal fluid of patients with neurological manifestations."

We have modified the Title according to the reviewer’s comment.

Reviewer 2 Report

Title: Hepatitis E virus quasispecies in the cerebrospinal fluid
This is a research paper where authors studied the hepatitis E virus quasispecies (HEV) in the blood and the CSF of 5 patients at the onset of their neurological symptoms (3 taken at the acute phase of the HEV infection and the other 2 taken during the chronic phase - 5 years after HEV diagnosis). Authors found for the first time evidence of HEV quasispecies compartmentalization in the CSF of the two patients during chronic infection concluding that prolonged infection in immunocompromised can lead to independent virus replication between the liver and the tissues producing viruses in the CSF. Very interesting and well-written study and I honestly have very little negative points to highlight. For all the above I advise minor revisions and provided a few suggestions below to improve the manuscript

Where authors state “The virus is also endemic in high-income countries, including Europe,” I would suggest “like those in Europe” or “including European countries”

On the goals paragraph authors underlined “cerebrospinal fluid”. Also, the acronym CSF had been described and used before so it should again here

I believe the PPR PCR conditions (and primer description) is lacking from the methods

Last sentence of methods; authors wrote “in the tree.3.2.”. please check 

Author Response

Where authors state “The virus is also endemic in high-income countries, including Europe,” I would suggest “like those in Europe” or “including European countries”

Wa have now modified this sentence as requested

On the goals paragraph authors underlined “cerebrospinal fluid”. Also, the acronym CSF had been described and used before so it should again here

I have now used the acronym through the text.

I believe the PPR PCR conditions (and primer description) is lacking from the methods.

The protocol is now added

Last sentence of methods; authors wrote “in the tree.3.2.”. please check 

We have deleted this tipo.

Reviewer 3 Report

The work is really interesting since it discusses the topic of variants or quasispecies of HEV in chronic infection, which is little known. The paper is very well written and very concise. In addition, the group is a world reference in HEV infection.

As a suggestion to the authors, I would discuss a little more the results obtained in other existing studies that have studied the HEV variants. An example is the article by Horvatits T et al. J Hepatol. 2021, in which the authors also describe HEV variants in chronic patients.

Author Response

We aggree with reviwer's comment. We have now mentionned the manuscript by Horvatits T et al. in the discussion section.